



**Insights into characteristics and formation mechanisms of secondary organic**
**aerosols in Guangzhou urban area**
**Miaomiao Zhai[1,3], Ye Kuang[1,3*], Li Liu[2,*], Yao He[1,3], Biao Luo[1,3], Wanyun Xu[4], Jiangchuan**
**Tao[1,3], Yu Zou[2], Fei Li[2,5], Changqin Yin[2,7], Chunhui Li[2], Hanbing Xu[6], Xuejiao Deng[2]**
[1] Institute for Environmental and Climate Research, Jinan University, Guangzhou, China.
[2] Key Laboratory of Regional Numerical Weather Prediction, Institute of Tropical and Marine
Meteorology, China Meteorological Administration, Guangzhou, 510640,China
[3] Guangdong-Hongkong-Macau Joint Laboratory of Collaborative Innovation for Environmental
Quality, Guangzhou, China.
[4] State Key Laboratory of Severe Weather & Key Laboratory for Atmospheric Chemistry, Institute of
Atmospheric Composition, Chinese Academy of Meteorological Sciences, Beijing, 100081, China
[5] Xiamen Key Laboratory of Straits Meteorology, Xiamen Meteorological Bureau, Xiamen, 361012,
China
[6] Experimental Teaching Center, Sun Yat-Sen University, Guangzhou 510275, China
[7] Shanghai Key Laboratory of Meteorology and Health, Shanghai Meteorological Bureau, Shanghai
200030, China
*Correspondence to: Ye Kuang (kuangye@jnu.edu.cn) and Li Liu (liul@gd121.cn)





**Abstract**

Emission controls have substantially brought down aerosol pollution in China, however, aerosol

mass reductions have slowed down in recent years in the Pearl River Delta (PRD) region, where
secondary organic aerosol (SOA) formation poses a major challenge for air quality improvement. In
this study, we characterized the roles of SOA in haze formation in urban Guangzhou City of the PRD
using year-long aerosol mass spectrometer measurements for the first time and discussed possible
pathways of SOA formations. On average, organic aerosols (OA) contribute dominantly (50%) to non-
refractory submicron aerosol mass (NR-PM$_1$). The average mass concentration of SOA (including by
less and more oxidized OA, LOOA and MOOA) contributed most to NR-PM$_1$, reaching about 1.7
times that of primary organic aerosols (POA, including hydrocarbon-like and cooking-related OA) and
accounting for 32% of NR-PM$_1$, even more than sulfate (22%) and nitrate (16%). Seasonal variations
of NR-PM$_1$ revealed that haze formation mechanisms differed much among distinct seasons. Sulfate
mattered more than nitrate in fall, while nitrate was more important than sulfate in spring and winter,
with SOA contributing significantly to haze formations in all seasons. Daytime SOA formation was
weak in winter under low oxidant level and air relative humidity, whereas prominent daytime SOA
formation was observed in fall, spring and summer almost on daily basis, suggesting for important
roles of photochemistry in SOA formations. Further analysis showed that the coordination of gas-phase
photochemistry and subsequent aqueous-phase reactions likely played significant roles in quick
daytime SOA formations. Obvious nighttime SOA formations were also frequently observed in spring,
fall and winter, and it was found that daytime and nighttime SOA formations together had resulted in
the highest SOA concentrations in these seasons and contributed substantially to severe haze
formations. Simultaneous increases of nitrate with SOA after sunset suggested the important roles of
NO$_3$ radical chemistry in nighttime SOA formations, and confirmed by continuous increase of
NO$^+$/NO$_2^+$ fragment ratio that related to measured particulate nitrate after sunset. Findings of this study
have promoted our understanding in haze pollution characteristics of the PRD and laid down future
directions on investigations of SOA formation mechanisms in urban areas of southern China that share
similar emission sources and meteorological conditions.


## 1 Introduction

Ubiquitous submicron aerosols in the atmosphere not only deteriorate human health and visibility, but also impact climate through interactions with solar radiation and clouds. Organic aerosols (OA) represent one of the most important and sometimes even dominant components (~10-90%) of $PM_1$ (aerosol particles with aerodynamic diameter less than 1 μm) in urban, rural and remote areas (Zhang et al., 2007;Jimenez et al., 2009). OA can either be emitted directly from emission sources or be formed through atmospheric reactions of volatile organic compounds, the former is referred to as primary OA (POA) and the latter is referred to as secondary OA (SOA). An increasing number of researches show that SOA account for a large fraction of OA worldwide (Zhang et al., 2007;Zhang et al., 2011), and even dominate in some cases (Kuang et al., 2020). The implementation of strict emission reduction policies has significantly improved the air quality of Pearl River Delta (PRD) region, which is a highly industrialized area of China, and the annual mean concentration of $PM_{2.5}$ (particulate matter with aerodynamic diameter less than 2.5 μm) has been brought down to less than 30 $μg/m^3$ (Xu et al., 2020). However, the reduction of $PM_{2.5}$ mass concentrations in PRD has slowed down substantially in recent years, which might be related to the significant increases in the proportion of secondary aerosols (Xu et al., 2020), especially for SOA. However, long-term observations that elucidate the sources and secondary formations of OA in this area is still missing.

Aerosol mass spectrometers are advanced on-line instruments that provide real time quantitative characterization of aerosol particle compositions (Jayne et al., 2000;Canagaratna et al., 2007;Jimenez et al., 2003). Positive matrix factorization (PMF) (Ulbrich et al., 2009) or a multilinear engine (ME-2) (Paatero, 1999;Canonaco et al., 2013) can be employed to further resolve different OA factors that are associated with different sources and formation mechanisms from the OA mass spectra. Although aerosol mass spectrometers have been widely used in China in recent years, most studies have been conducted in specific periods due to its high cost and maintenance (He et al., 2011;Chen et al., 2021;Qin et al., 2017), resulting in few long-term characterizations of the mass concentrations and chemical compositions of $PM_1$. The design of Aerosol Chemical Speciation Monitor (ACSM) has improved this problem to some extent (Ng et al., 2011;Sun et al., 2015). Based on 2-year ACSM measurements, Sun et al. (2018) investigate the distinct characteristics of $PM_1$ compositions among different seasons in Beijing urban area. Canonaco et al. (2021) also performed a long-term source





apportionment on a 1-year ACSM dataset from downtown Zurich. Many other studies also have
successfully applied the ACSM in the monitoring organic aerosols in various regions (Sun et al.,
2012;Sun et al., 2013;Sun et al., 2014;Sun et al., 2016;Fröhlich et al., 2013;Allan et al., 2010;Zhang et
al., 2012;Xu et al., 2015a;Zhou et al., 2020;Huang et al., 2014;Hu et al., 2016;Via et al., 2021), however
long-term measurements are still relatively scarce, and remain missing in urban areas of the PRD
region.
Guo et al. (2020) found that OA played a dominant role in $PM_1$ during winter in Guangzhou, a
mega city of the PRD, and the results of OA source apportionment emphasized the dominance of SOA.
Qin et al. (2017) and Huang et al. (2011) also reported similar results during autumn and winter in
Guangzhou. In fact, the compositions, sources, and evolution processes differ much among seasons
due to changes in emission sources and meteorological conditions (Li et al., 2015). Thus, long-term
characterizations that cover measurements of different seasons are urgently needed to gain insights
into emission sources and formation mechanisms of OA, thereby helping to address the challenge of
fine particulate matter pollution mitigation in the PRD region.
In this study, we performed a year-long continuous measurement of non-refractory submicron
aerosols (NR-$PM_1$) with an ACSM in urban Guangzhou from September 2020 to August 2021 to
characterize POA sources and investigate SOA formation mechanisms in different seasons.

## 2 Experimental methods

### 2.1 sampling site and measurements

A quadrupole-Aerosol Chemical Speciation Monitor (Q-ACSM) was deployed to continuously
measure nonrefractory $PM_1$ (NR-$PM_1$) species including OA, sulfate ($SO_4$), nitrate ($NO_3$), ammonium
($NH_4$), and chloride (Cl) from September 2020 to August 2021 at an urban site located in Haizhu
wetland park of Guangzhou, which is surrounded by commercial streets and residential buildings,
however, with a distance of at least 1 km (Liu et al., 2022). Therefore, measurements at this site are
representative of the pollution characteristics of Guangzhou urban area. More detailed descriptions
about the sampling site and the ACSM measurements could be referred to Liu et al. (2022) and Ng et
al. (2011), respectively. An AE33 aethalometer (Drinovec et al., 2015) set up with a flow rate of 5





L/min was separately operated downstream of a PM$_{2.5}$ inlet (BGI SCC 1.829) to measure aerosol
absorptions, from which black carbon (BC) mass concentrations in winter and early spring. In addition,
mass concentrations of PM$_{2.5}$ and trace gases such as nitrogen dioxide (NO$_2$), ozone (O$_3$), carbon
monoxide (CO) and sulfur dioxide (SO$_2$) were acquired from the publicly available datasets of the
China National Environmental Monitoring network (http://www.cnemc.cn/en/), which includes a site
located within 5 km distance to our observation site. Measurements of meteorological parameters such
as temperature, wind speed and direction (WS and WD), and relative humidity (RH) were made by an
automatic weather station (Li et al., 2021). Aerosol liquid water content (ALWC) was predicted with
the ISORROPIA-II thermodynamic model in reverse mode under metastable assumption (Guo et al.,
2017) with aerosol chemical compositions measured by Q-ACSM as inputs, with more details in
Supplement Sect.S2.
**2.2 Q-ACSM data analysis**
The Q-ACSM data were processed using ACSM standard data analysis software (ACSM Local
1.5.10.0 Released July 6, 2015) written in Igor Pro (version 6.37). The composition-dependent
collection efficiency (CE) parameterization scheme proposed by Middlebrook et al. (2012) was chosen
to determine the mass concentrations of NR-PM$_1$ species which was also detailed in Liu et al. (2022).
Relative ionization efficiencies (RIEs) of 5.15 and 0.7 were adopted for ammonium and sulfate
quantifications which were calibrated using 300 nm pure NH$_4$NO$_3$ and (NH$_4$)$_2$SO$_4$ while the default
RIEs of 1.4, 1.1 and 1.3 was used for organic aerosol, nitrate and chloride, respectively. Moreover, we
also compared the mass concentrations of NR-PM$_1$ with PM$_{2.5}$ to ensure the validity of ACSM data
during the whole study. As shown in Fig. S1 of the supplement, the measured NR-PM$_1$ correlates
highly with PM$_{2.5}$ acquired from the nearest (about 5 km) Environmental Protection Agency site (R$^2$ =
0.71), and the average ratio of NR-PM$_1$/PM$_{2.5}$ is 0.77 (±0.36).





Unconstrained Positive matrix factorization (PMF) was performed on OA mass spectra of the entire
year-long dataset. For the two-factor solution, the POA factor peaked in the evening with low O/C
(~0.28) and an oxygenated OA (OOA) factor peaks in the afternoon with high O/C (~0.88) can be well
resolved (Fig.S2), demonstrating the markedly different influences of primary emissions and SOA
formations on diel aerosol mass concentrations. However, PMF-ACSM analysis of mass spectra of

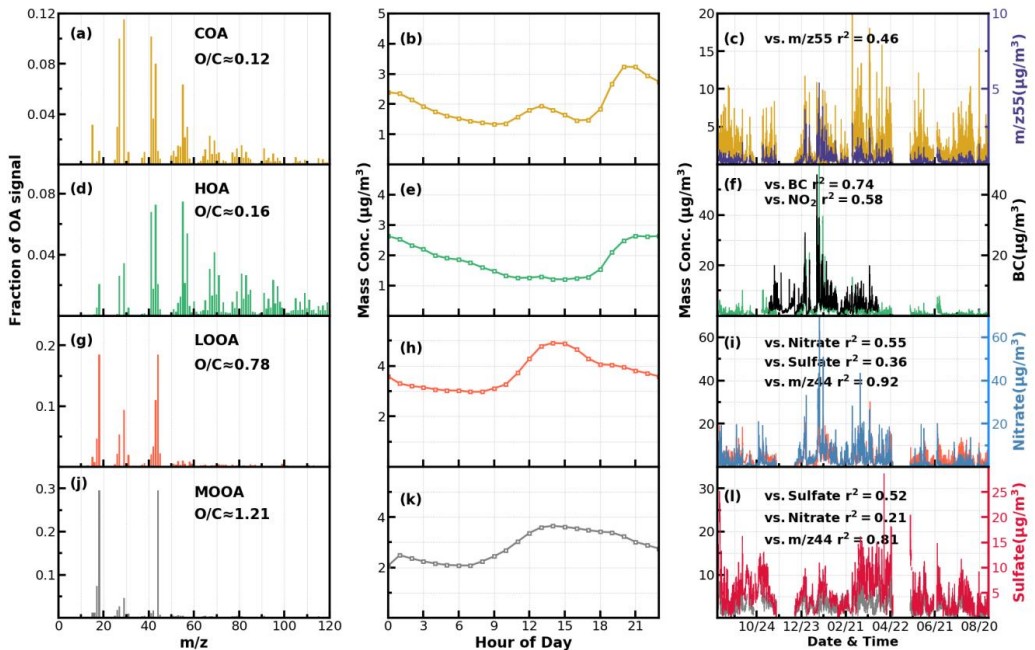

**Figure 1.** Mass spectral profiles, diurnal cycles and correlations with external data of COA**(a-c)**, HOA**(d-f)**, LOOA**(g-i)** and MOOA**(j-l)** from ME2-ACSM analysis for the entire year.

OA measured by unit mass resolution instruments still faced some uncertainties to further resolve
potential POA or SOA components due to its rotational indeterminacy. For example, traffic-related
hydron-carbon like organic aerosols (HOA) was uneasily to separate from cooking-related organic
aerosols (COA) and there was also great uncertainty in distinguishing SOA with different degrees of
oxidations (Sun et al., 2012;Sun et al., 2013;Zhang et al., 2015). Therefore, an improved source
apportionment technique called Multilinear Engine (ME-2) was further used to resolve better sources
of POA and SOA (Paatero, 1999;Canonaco et al., 2013;Guo et al., 2020). Previously, both Guo et al.
(2020) and Liu et al. (2022) demonstrated that during both autumn and winter seasons of Guangzhou
urban areas, POA was mainly composed of HOA, which is mostly associated with traffic emissions



and COA, and SOA could be resolved into less oxidized and more oxidized organic aerosols (LOOA
and MOOA). The number selecting test using unconstrained PMF analysis (Fig.S3) also showed that
four-factor solution likely be the best choice. Therefore, we had chosen 4 factors for ME-2 analysis
with the *a* value of ME-2 ranges from 0.1 to 0.5, and constrained the HOA and COA profiles with
HOA and COA profiles reported in Liu et al. (2022) as priories considering the following three reasons:
(1) The used instrument of this study is the same one of Liu et al. (2022); (2) the COA profile reported
in Liu et al. (2022) was determined during the period when both COVID-19 silence-action and festival
spring occurred when cooking activities grew and traffic activities almost vanished thus COA shall
dominated over HOA, more details about the method please refer to Liu et al. (2022); (3) Resolved
variations of HOA and COA are well explained by external datasets such as correlations of HOA with
black carbon reached 0.79. The four-factor solution using the ME-2 technique with *a*=0.2 was obtained
and shown in Fig.1. The resolved HOA and COA are summed as POA, resolved LOOA and MOOA
are summed as SOA, and the comparison with those resolved by the PMF is shown in Fig.2. ME-2
analysis generally reproduced both the diurnal variations as well as absolute mass concentrations of
POA and SOA during different months well. To explore the consistency of resolved factors using the
entire year-long dataset or only using seasonal dataset when performing ME-2 analysis, we performed
individual ME-2 runs for each season. Results showed that factors resolved in each season using
seasonal datasets as inputs of ME-2 are generally consistent with those resolved from year-long dataset
(Fig.S4-S7). Therefore, factors resolved using the entire year-long dataset as input of ME-2 were used
for further investigations and this also guaranteed consistency of factors for comparisons among
seasons.



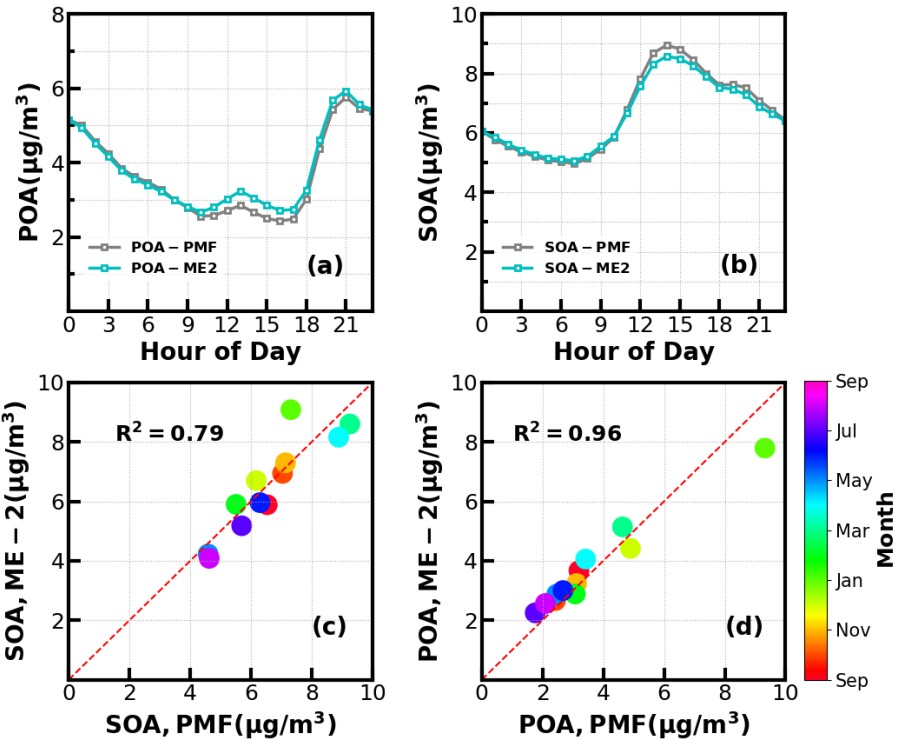

**Figure 2.** **(a)** and **(b)** Diurnal variations of POA and SOA concentrations from ME-2 and PMF; **(c)** and **(d)** Scatter plots between monthly average POA and SOA concentrations from ME-2 and PMF.

The mass spectrum of COA deconvolved in this work was characterized by a high m/z 55-to-57
ratio of 2.12, which was the same with the one reported by Guo et al. (2020), and close to the m/z 55-
to-57 ratio range of 2.2-2.8 reported by Mohr et al. (2012) for COA. Similar to previous studies (Guo
et al., 2020;Sun et al., 2013), the concentration of COA was well correlated ($R^2$=0.46) with m/z 55.
The O/C ratio of 0.12 for COA revealed that it was less oxidized than HOA (O/C=0.16) during the
whole year in Guangzhou, which was contrary to Sun et al. (2011). As shown in Fig.3, the diurnal
profile of COA presented two typical peaks during the entire year with a noontime peak during 13:00
- 14:00 LT and an evening peak during 20:00 - 21:00 LT, which were associated with noon and evening
cooking activities. It was noteworthy that the nighttime peak concentration of COA was very close to
that of noontime in summer, while the evening peak of COA was significantly higher than that of
noontime in other three seasons. The ratio of evening COA peak to that of the noontime was 1.7 in fall,
and was 1.6 in spring. In particular, the evening COA peak was nearly 4 times that of noontime in
winter due to the relatively insignificant noontime peak during this period, which might be associated



with the lock down and spring festival in winter which resulted in less noontime activities. Similar
conclusions could be found in Sun et al. (2018). More frequent cooking activities at night such as the
Chinese habit of eating midnight snacks, shallower boundary layer that inhibited diffusion of pollutants,
and the lower temperature at night which facilitated semi-volatile compounds from cooking emissions
to partition into particles resulted in the higher peak concentration at nighttime than at noon (Guo et
al., 2020).

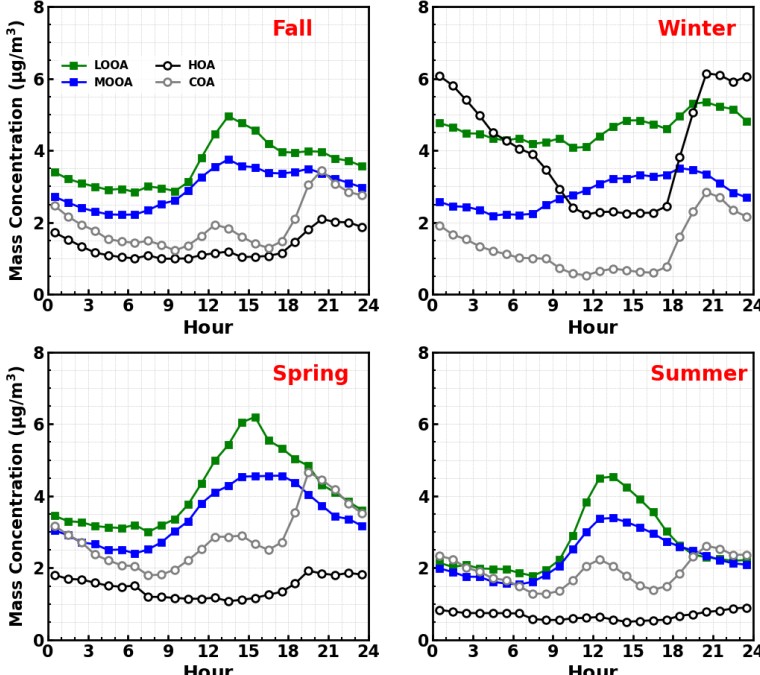

**Figure 3.** Diurnal profiles of HOA, COA, LOOA and MOOA in spring (March to May), summer (Jnue to August), Fall (September to November), Winter (December to February).

The mass spectrum of HOA (Fig.1b) was characterized with the $C_nH_{2n-1}^+$ (m/z = 27, 41, 55, 69)
and $C_nH_{2n+1}^+$ (m/z = 29, 43, 57, 71) ion species. The concentration of HOA had a good correlation
with that of primary BC emission ($R^2$=0.74), and also correlated well with that of $NO_2$ ($R^2$=0.58),
indicating considerable impacts of traffic emissions on the HOA mass loading. As shown in Fig.3,
except for summer, HOA increased significantly after sunrise especially in winter, however, began to
decrease in the late evening. HOA was significantly higher during nighttime than during daytime in
all seasons especially in winter, however, was not obvious in summer. HOA mass concentration peaks
around 20:00 LT were attributed to traffic emissions during the nocturnal rush hours. However, the





continuously high concentrations of HOA after 20:00 until 02:00 of the next day might have resulted
from heavy-duty vehicles with daytime traffic restrictions in Guangzhou (Guo et al., 2020;Qin et al.,

2017).

Two OOA factors were characterized with high O/C ratio, LOOA with O/C of 0.78 and MOOA
with O/C of 1.2, suggesting high oxidation degrees of SOA factors in Guangzhou urban area, especially
that of MOOA. MOOA and LOOA shared similar diurnal profiles regardless of seasons, with MOOA
showed higher correlations with sulfate and LOOA showed higher correlations with nitrate. MOOA
and LOOA increased together in fall from 09:00 LT until 14:00 LT reached a maximum of 3.7 μg/m$^3$
for MOOA and 5 μg/m$^3$ for LOOA, followed by a gradual decrease in SOA concentrations and then
remained relatively flat. The diurnal profiles of SOA in spring and summer were relatively similar to
those in fall, however, more remarkable decreases of SOA from afternoon to midnight were observed
in spring and summer. This is because SOA sometimes increased after sunset in autumn, which was
even more prominent in winter, where LOOA and MOOA would first increase for a while after sunset
and then begun to decrease. However, weaker daytime SOA formation was observed in winter.
**3 Results and discussion**
**3.1 The largest contribution of secondary organic aerosols in NR-PM$_1$**
Time series of the meteorological parameters (including RH, WS and WD), the mass
concentrations of NR-PM$_1$ and PM$_{2.5}$, chemical compositions of NR-PM$_1$, trace gases and four
resolved OA factors are shown Fig.S8. It shows that emission source intensities and meteorological
variables changed dramatically among seasons. Hourly NR-PM$_1$ mass concentrations ranged from near
zero to 177 μg/m$^3$ with an average of 21 μg/m$^3$. From October to February, northerly winds prevailed
and average NR-PM$_1$ was relatively higher than that from February to September (26 vs 19 μg/m$^3$),
which were associated with relatively lower boundary height during cold seasons and northern winds
brought polluted continental air mass. While during warm seasons of Guangzhou (March to
September), south-easterly wind prevailed, which brought cleaner air mass from the ocean and the
boundary layer height was higher due to more surface heating. Monthly variations of PM$_{2.5}$ are shown
in Fig.3a, PM$_{2.5}$ in summer was lowest and around 16 μg/m$^3$ from May to August which were likely





associated with the prevalence of rainy conditions in summer and possible higher boundary layer
height. January was the month with highest $PM_{2.5}$ mass concentrations with an average of 49 μg/m$^3$,
which was consistent with the fact that winter usually experienced the worst air pollutions due to the
stagnant air conditions.

The average mass contributions of different components to NR-PM$_1$ during the entire year and

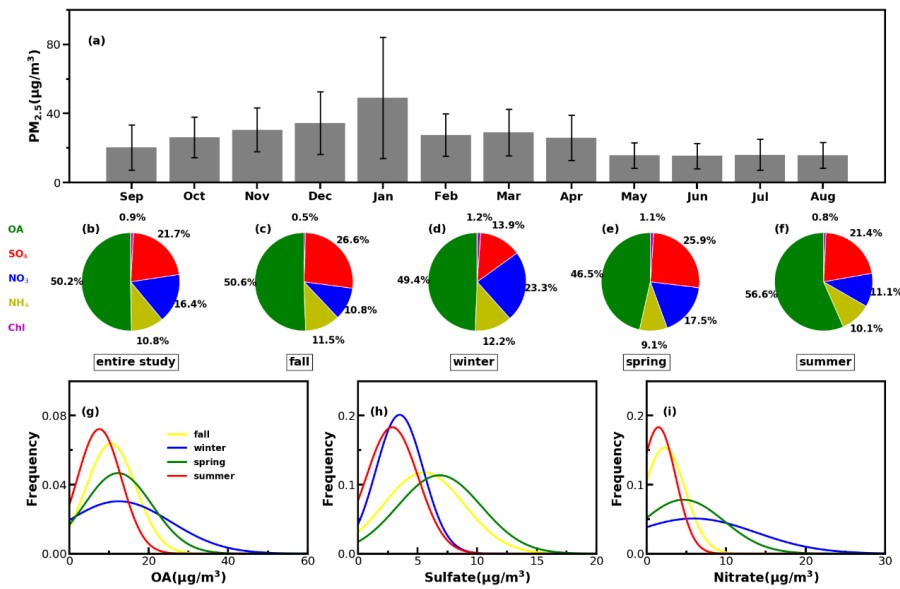

**Figure 3. (a)** Monthly average $PM_{2.5}$ mass concentrations from September of 2020 to August of 2021; **(b)-(f)** The average mass fractions of the chemical components in NR-PM$_1$ of the entire year and different seasons; **(g)-(i)** Probability distributions of OA, sulfate (SO$_4$) and nitrate (NO$_3$) in different seasons.

among different seasons are shown in Fig.3b-3f. On average OA contributed about 50% to NR-PM$_1$
with the highest contribution in summer that reached near 57% and lowest contribution in spring of
about 47%. The second largest contributor was sulfate, which on average contributed about 22%, and
more than 20% in spring, summer and fall. However, the contribution of nitrate to NR-PM$_1$ (23%)
exceeded that of sulfate (14%) and became the second major component after OA in winter, consistent
with the results of Guo et al. (2020) for pollution periods in winter of Guangzhou. The probability
distributions of mass concentrations of OA, sulfate and nitrate are shown in Fig.3g-3i. Both OA and
nitrate were distributed in wide ranges during winter and shared similar shape of probability
distribution, with OA increasing gradually from summer to winter and then reducing in the spring.
Sulfate shared similar magnitudes in summer and winter, and differed much from those in spring and



fall that had higher sulfate concentrations and varied in a wider range. Nitrate in summer and fall were
relatively lower in summer and fall, however, had much higher concentrations in spring and winter.

As shown in Fig.4a, average OA concentrations of different months ranged from about 7 μg/m$^3$

to 17 μg/m$^3$ with the peak in January and the lowest in August, and the variations of OA mass
concentration in winter and spring were much larger than those in summer and autumn. Monthly

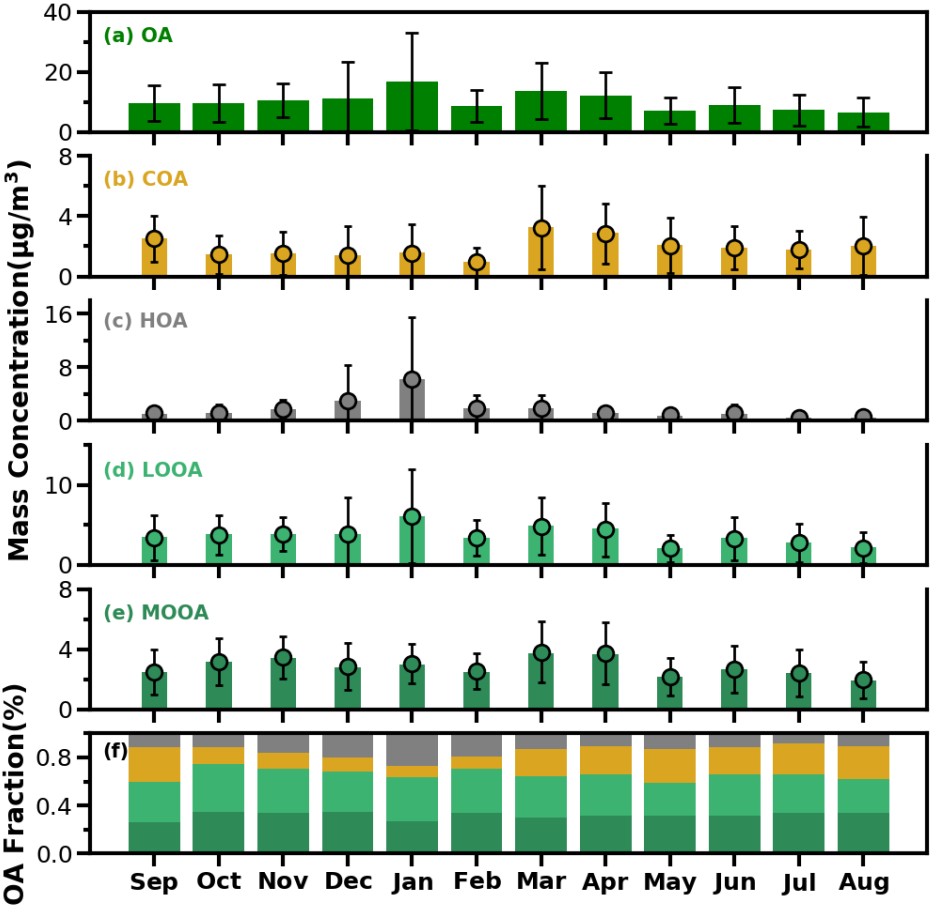

**Figure 4.** The bar plots of monthly average mass concentrations of OA, COA, HOA, LOOA and MOOA from **(a)** to **(e)** and mass fractions of OA factors in OA **(f)**.

variations of mass concentrations of the four resolved factors are shown in Fig.4b-e, and contributions
of the four OA factors to OA are shown in Fig.4f. In general, HOA remained lower than 2 μg/m$^3$ in
most months, however, as the cold season approached from November, the monthly average OA
increased substantially from about 2 μg/m$^3$ to near 6 μg/m$^3$. The much lower temperature and





accumulation favorable meteorological conditions likely had resulted in the substantial increase of HOA. The obviously higher concentrations of $NO_2$ even under lower wintertime $O_3$ concentrations (Fig.S8) implied that more traffic emissions in winter might also had contributed to the substantial HOA increase. Compared with HOA, the seasonal variations of COA were less pronounced. The monthly average concentration of COA in warm months (February to October) was higher than those in cold months (October to January). The lowest monthly average concentration of COA was about 1 µg/m$^3$ which occurred in February when the contribution of COA to OA was near its lowest of about 9%. Overall, the contributions of COA to OA were higher than that of HOA during warm months, however, lower than that of HOA in relatively cold months especially in winter, and contributed about 19% of OA during the whole year, which was close to that of HOA (18%). These results highlight the significant contributions of POA to OA in Guangzhou urban area, however, contributions of emission sources differed much among cold and warm seasons.

SOA (MOOA+LOOA) contributed more than 60% to OA in all months, reached beyond 70% in October and February, and made up on average 63% of OA in the entire year. As shown in Fig.4(e-f), LOOA exhibited stronger seasonal variations than MOOA, with monthly average mass concentrations of LOOA varying between 2.6 to 6.1 µg/m$^3$ and monthly average MOOA concentration ranging from 2 to 3.8 µg/m$^3$. The LOOA mass concentration peaked in the most polluted month of January, suggesting that significant contributions of LOOA formation to severe haze pollution in winter. The contribution of LOOA to OA ranged from 27% to 39% with an average of 34%, and the contribution of MOOA to OA ranged from 26% to 35% with an average of 32%. Overall, the average mass concentration of SOA was about 1.7 times that of POA for the whole year, and SOA accounted for about 32% of NR-PM$_1$, which was higher than those of sulfate and nitrate, demonstrating the largest contribution of SOA to NR-PM$_1$.

**3.2 Significant contributions of secondary organic aerosols to haze formations in all seasons**

Investigations on contribution variations of aerosol compositions under different aerosol pollution levels are helpful for understanding mechanisms of haze formations, and results in four seasons are presented in Fig.5. The chemical composition of NR-PM$_1$ under different pollution levels differ much among seasons. In fall, as demonstrated by variations of mass concentrations of aerosol compositions





under different pollution levels shown in Fig.5, pollution conditions in fall were dominantly controlled by secondary formations of sulfate and SOA, accumulation of primary aerosols and nitrate formation had relatively smaller impacts. With respect to mass fractions variations, contributions of aerosol components differed much among different pollution levels. The fraction of OA decreased rapidly from 67% to 50% when the mass concentration of NR-PM$_1$ gradually increased to 15 μg/m$^3$, while the contribution of sulfate increased substantially from 17% to 30%, and the contribution of nitrate remained relatively stable at about 10%. When NR-PM$_1$ further increased, OA contribution remained relatively flat for NR-PM$_1$ below about 50 μg/m$^3$. Accordingly, the contribution of SO$_4^{2-}$ decreased to ~18%, and the contribution of nitrate substantially increased from ~10 % to 21%. After that, OA contribution decreased rapidly to about 40% and then remained stable for NR-PM$_1$ >50 μg/m$^3$. However, the contribution of sulfate began to increase, and the highest contribution could account for 30%, while the contribution of nitrate began to decline gradually to 12%. In addition, the SOA contributed dominantly to OA (>60%) for NR-PM$_1$ > 15 μg/m$^3$ and even reached near 70% for NR-PM$_1$ > 35 μg/m$^3$, suggesting the dominant role of SOA in OA accumulations in haze events during fall.

In winter, haze formations are mostly associated with POA accumulations, SOA and nitrate formations, with nitrate formation playing the most important role, since it is also accompanied by ammonium formation, while sulfate formation was weak in winter. The fraction of OA increased gradually with the increase of NR-PM$_1$ concentration for NR-PM$_1$ < 90 μg/m$^3$ and reached the maximum of 60%, while the contribution of nitrate also showed a small increase from 21% to 26%. Under aggravating pollution, OA contribution fluctuated, however, showed a decreasing trend from 60% to ~40%. Meanwhile, the nitrate contribution showed an increasing trend from 26% to ~40%, which was similar to that of OA. Sulfate contribution decreased with the increase of NR-PM$_1$ concentration for NR-PM$_1$ < 100 μg/m$^3$ and then remained at about 6% as NR-PM$_1$ increases. In addition, the POA contribution increased about 25% to 50% for NR-PM$_1$ < 100 μg/m$^3$. Overall, the increase of nitrate, POA and SOA together had resulted in severely polluted conditions in winter. The substantial contribution of POA to severe haze demonstrates that meteorological conditions unfavorable for the pollutant diffusion together with the substantial contributions of secondary nitrate and SOA formations have resulted in the most severe haze pollutions among the year. Especially, HOA contribution to OA increased from 17% to 52% when NR-PM$_1$ concentration was less than 140 μg/m$^3$,

suggesting the significant role of traffic emission accumulation during severe haze pollution, which

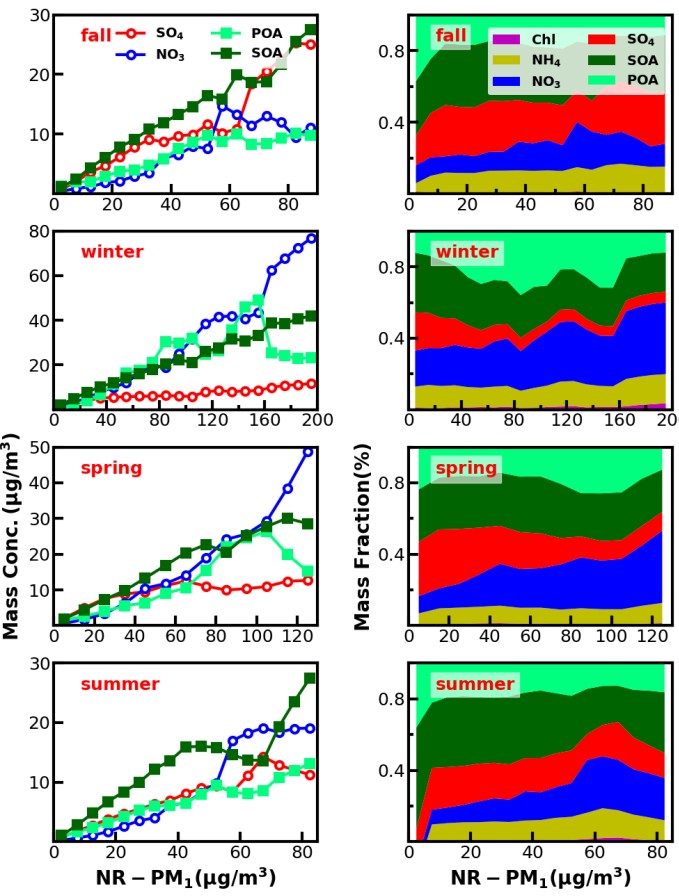

**Figure 5.** Left panels show absolute mass concentration variations of aerosol compositions under different NR-PM$_1$ levels, right panels show mass fractions of chemical components as a function of NR-PM$_1$.

was consistent with results of Yao et al. (2020).
In spring, haze pollutions were mostly associated POA accumulation and secondary formations of
nitrate and SOA, especially that of nitrate. The contribution of OA decreased from 51% to 44% as NR-
PM$_1$ mass concentration increased when NR-PM$_1$ mass concentration was less than 50 μg/m³. When
the mass concentration of NR-PM$_1$ reached about 105 μg/m³, the fraction of OA reached a maximum
of 55%, and then decreased to about 37%. The most noticeable characteristic was the increase of nitrate
contribution (from 10% to 40%) and decrease of sulfate contribution (32% to 10%) as the NR-PM$_1$
increased. In summer, secondary aerosol formations contributed dominantly to haze formations, with



POA contribution to NR-PM$_1$ was about 20% in most conditions. The overall contribution of OA
gradually decreased from near 60% to 35% as the mass concentration of NR-PM$_1$ increased for NR-
PM$_1$ concentration < 60 μg/m$^3$ which was markedly different with those in other seasons, however
increased to 49% as the NR-PM$_1$ concentration increased further. The contribution of sulfate decreased
from 25% to 13% and the contribution of nitrate increased from 9.0% to 31% with the increase of NR-
PM$_1$ concentration for NR-PM$_1$ concentration < 60 μg/m$^3$. While the OA was dominated by SOA under
most conditions (about 60%).
Overall, haze formation mechanisms differed much among distinct seasons. Sulfate mattered more
than nitrate in   fall, while nitrate mattered more than sulfate in spring and winter, however, SOA
contributed significantly to haze formations in all seasons.

**3.3 Discussions on SOA formation mechanisms**

SOA can be formed through condensation of oxidized gas-phase organic vapors during the
oxidation of volatile organic compounds (VOCs), this type of formed SOA was usually referred to as
gasSOA (Kuang et al., 2020). SOA can also be formed in the aqueous phase through the further
oxidation of dissolved VOCs which are usually products of gas-phase oxidation of VOCs, this type of
SOA was usually referred as aqSOA (Kuang et al., 2020). As shown in Fig.3, both LOOA and MOOA
mainly increased after sunrise, highlighting important roles of photochemistry in SOA formations.
However, as demonstrated by Kuang et al. (2020), the daytime SOA formation could be either result
from gas-phase photochemistry and subsequent condensation (gasSOA), or the result of gas-phase
VOCs transformations with subsequent aqueous reactions (aqSOA). Especially since the PRD region
is characterized by both active photochemistry due to strong solar radiation in subtropical regions and
high relative humidity (annual average RH of ~75%), both photochemistry and aqueous phase
reactions might play significant roles in SOA formation, however, this aspect was not explored before.



Considering the frequent co-increase of MOOA and LOOA, they were grouped together as SOA

for further investigations on their formation. SOA formation cases in four seasons were identified, the

start time and lasting hours of their occurrences, as well as associated SOA levels are shown in Fig.6.

Note that the identification of SOA formation cases has not considered the dilution effect of the lifting

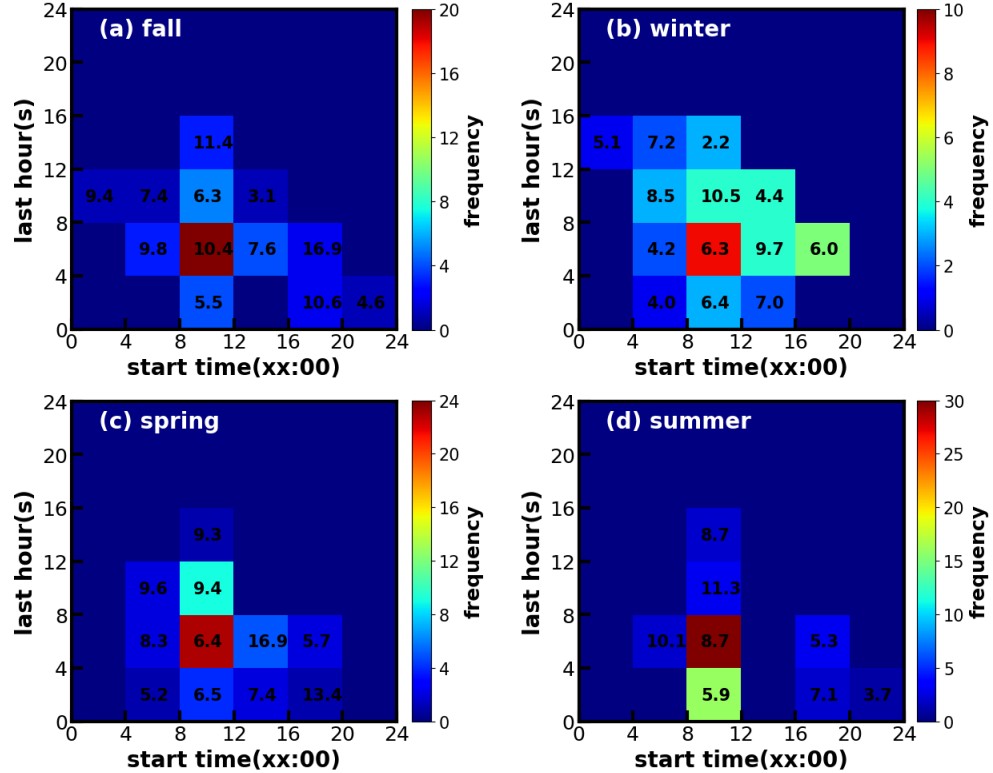

**Figure 6:** Time frequency diagrams of SOA increase events in **(a)** fall, **(b)** winter, **(c)** spring and **(d)** summer. X-axis represent start time of SOA increase, and y-axis represents the lasting hours of SOA increase events. The color scales indicate the number of occurrences. The values in the grid are the average SOA concentration during the SOA increase case.

daytime boundary layer height and was only based on the absolute mass concentration variations.

Therefore, this method has neglected some SOA formation cases that were masked by evolutions of

the boundary layer, and the identified cases represent active SOA formation events that overcame

dilution effects, which might be more suitable for further SOA formation investigations due to strong

SOA formation signals. It shows that in all seasons, the SOA formation happened most frequently

during daytime, starting in the morning and lasting about 4-8 hours. Especially, in spring, summer and



fall, the daytime SOA formation almost happened everyday (Fig.S5-7), even if strong daytime
boundary layer evolutions could be expected in these seasons due to strong surface solar heating, and
resulted in the afternoon SOA mass concentration peaks in these seasons (Fig.3). However, highest
SOA concentrations did not appear in the seasons with the most frequent morning to afternoon
increases. Taking SOA formation cases in spring as an example, if the SOA increase started in the
morning, more than 8 hours duration will result in significant higher SOA concentration. These cases
started in the afternoon and lasted 4-8 hours would result in highest SOA concentration in spring. The
SOA formation cases starting in the morning, however, only lasting within 4 hours, happened
frequently in summer while less in spring and fall, suggesting that the absolute SOA mass
concentration increase was more often stopped by strong boundary layer mixing in summer, which
was consistent with the solar heating characteristics. The highest SOA in fall and winter were
associated with the continuous increase of SOA after sunrise, suggesting that coordination of daytime
and nighttime SOA formation together had resulted in the highest SOA concentrations in fall and winter.
To dig deeper into possible mechanisms behind the active daytime SOA formations throughout
the year, we investigated relationships between SOA formation rates and both $O_3$ as well as aerosol
liquid water content (ALWC) for the most frequent morning to afternoon SOA increase cases. Without
considering the dilution effect of rising boundary layer, the daytime apparent growth rates of SOA
varied from 0.2 to 4.4 μg m$^{-3}$ h$^{-1}$ (Fig.7). Note that the SOA growth rates was calculated on the basis
of observations of the first four hours for each SOA increase case to reduce impacts of boundary layer
dilution effects. Some previous studies used variations of CO concentrations to partially correct for
boundary layer dilution effects, however this method would fail in sites with strong CO emissions
(Kuang et al., 2020). The SOA growth rates and were highly correlated to $O_3$ formation rates (r=0.7)
as shown in Fig.7. However, this result only proved the important role of photochemistry in SOA
formations. The apparent SOA growth rates showed positive but much weaker correlation with the
average $O_3$ concentration during the period of SOA the increase (r=0.38), demonstrating that oxidant
level was likely not the controlling factor for SOA formation, although $O_3$ alone did not represent the

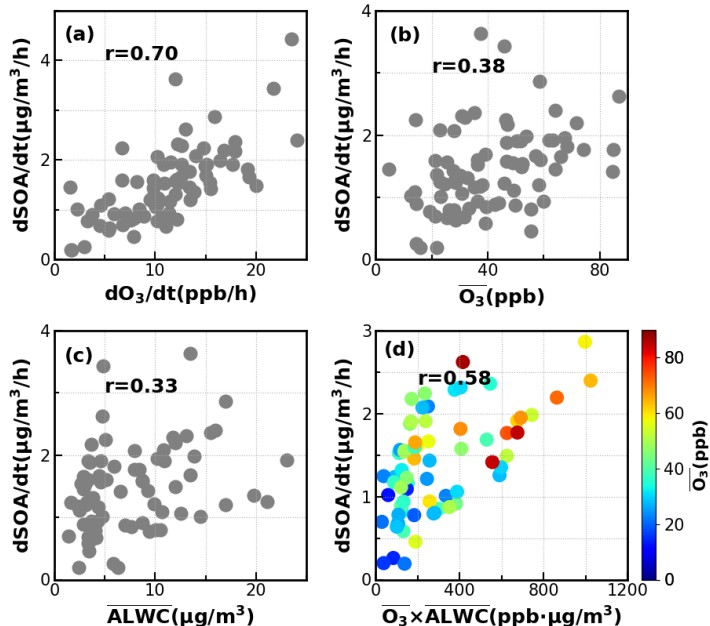

**Figure 7.** Relationships between SOA daytime formation rates with corresponding **(a)** $O_3$ formation rate; **(b)** average $O_3$; **(c)** average ALWC ($\mu g/m^3$) and **(d)** combination of averaged $O_3$ and averaged ALWC.

variations of oxidation levels and other sources such as HONO photolysis (Yu et al., 2022) also
contribute to OH radicals and is a typical oxidant in daytime photochemistry. To investigate the
possible roles of aqueous reactions in SOA formation, the relationship between apparent SOA rates
and corresponding average ALWC were also investigated, and a positive but weak correlation was
found (r=0.33). More importantly, the correlation coefficient between apparent SOA growth rates and
the variable of average ALWC multiplying by average $O_3$ would be much higher (r=0.58, Fig.7d),
suggesting that the coordination of gas-phase photochemistry and further aqueous reactions had likely
resulted in the rapid daytime SOA formations.





Besides the daytime SOA formation associated with photochemistry, dark transformations of
VOCs that involve nighttime gas-phase and aqueous phase reactions might also result in efficient SOA
formations. As shown in Fig.6, continuous increases of SOA were also frequently observed after sunset

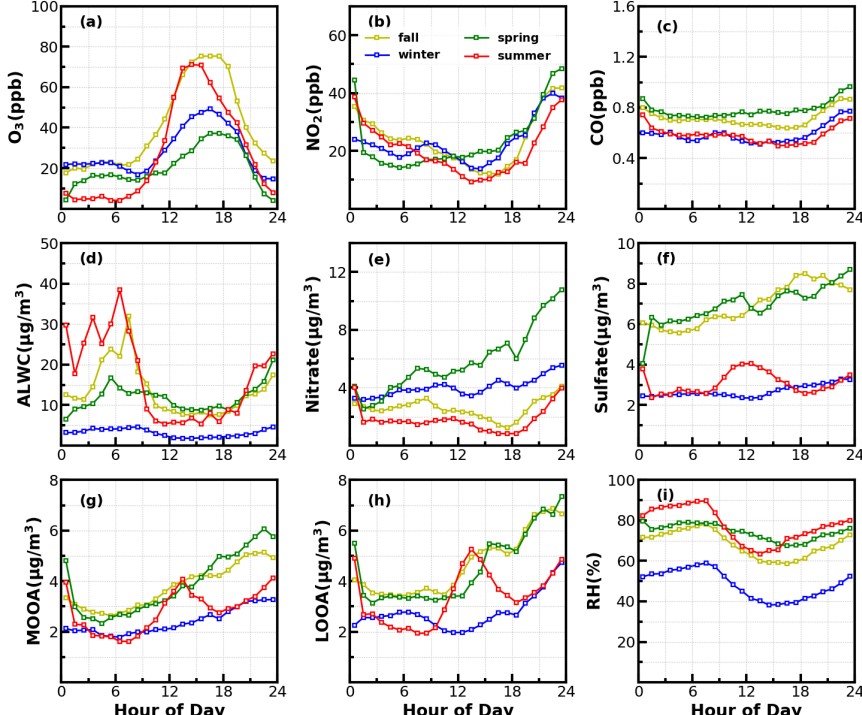

**Figure 8.** Average diurnal variations of **(a)** $O_3$; **(b)** $NO_2$; **(c)** CO; **(d)** ALWC; **(e)** nitrate; **(f)** sulfate; **(g)** MOOA; **(h)** LOOA and **(i)** RH for identified days with nighttime SOA increases.

in spring (17 days), fall (18 days) and winter (20 days) with sporadic occurrence in summer, and the
coordination of daytime and nighttime SOA formations together have resulted in the highest SOA
concentrations in fall and winter which were associated with severe haze pollutions as demonstrated
above. Average diurnal profiles of $O_3$, $NO_2$, CO, RH, ALWC, nitrate, sulfate, LOOA and MOOA for
cases with co-increases of LOOA and MOOA after 18:00 in different seasons are shown in Fig.8. On
average, SOA usually showed decreases during nighttime (Fig.3) due to transport of air mass from
cleaner suburban regions. The average wind speed was 1.7 m/s from 18:00 to 23:00 LT for identified
nighttime SOA increase cases and was obviously lower than the corresponding average wind speed of
2.3 m/s, suggesting the more stagnant air mass tended to favor the nighttime SOA increases. However,


the nighttime 5h back trajectories shown in Fig.S9 demonstrated that the nighttime replacement of

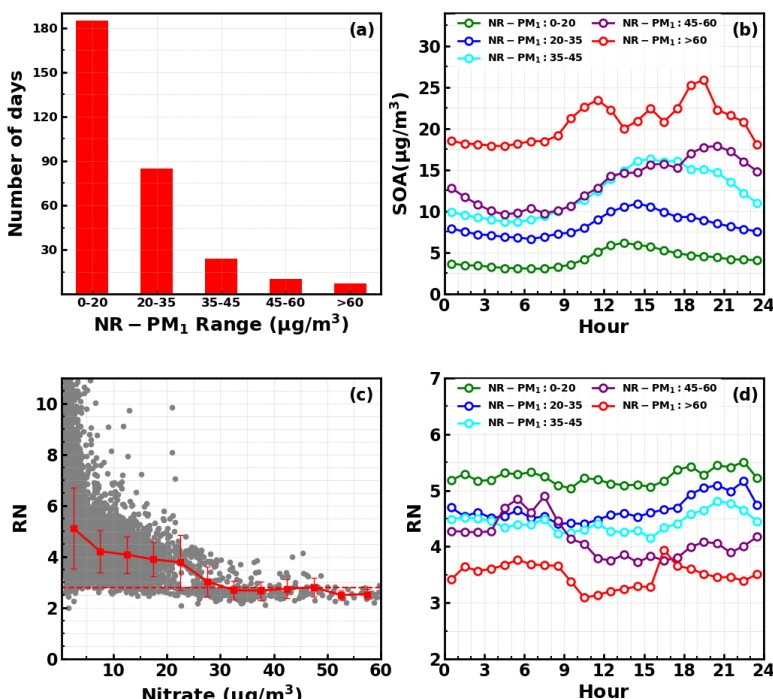

**Figure 9.** **(a)** Number of days in different daily average NR_PM$_1$ ranges; **(b)** Diurnal profiles of SOA under different NR_PM$_1$ ranges; **(c)** Variations NO+/NO2+ (RN) as a function of measured nitrate, horizontal dashed line corresponds to RN of 2.8, red markers and bars represents averages and standard deviations; **(d)** Diurnal profiles of RN under different NR_PM$_1$ ranges.

surrounding suburban cleaner air mass still prevailed, therefore the continuous increases of SOA
suggested that nighttime SOA formation occurred on a regional scale. The increases of LOOA and
MOOA were accompanied with obvious nitrate formation in all seasons as well as slight increases of
sulfate, further indicating for regional scale nighttime secondary aerosol formations during these
nighttime SOA formation events. Except for summer, continuous increase of SOA from the morning
to nighttime confirmed that the coordination of daytime and nighttime SOA formations had contributed
to haze formations. Number of days for daily average NR-PM$_1$ ranges of 0-20, 20-35, 35-45, 45-60
and >60 μg/m$^3$ were 185,85,24,10 and 7, respectively (Fig.9a). All cases with daily average NR-PM$_1$
higher than 45 μg/m$^3$ occurred in fall, winter and spring. The corresponding average diurnal variations
of SOA for these relatively severe conditions shown in Fig.9b confirmed further that the coordination



of daytime and nighttime SOA formations had contributed to severe haze formations in Guangzhou
urban area.

The $NO_3$ radical formed through the rection between $NO_2$ and $O_3$ is the typical nighttime oxidant.

Results of Rollins et al. (2012) and Kiendler-Scharr et al. (2016) revealed that $NO_3$ oxidation of VOCs
would contribute substantially to nighttime SOA increase. As shown in Fig.8a, after sunset, the $O_3$
concentration decreased quickly, however, remained substantially higher than zero, accompanied was
the remarkable increases of $NO_2$ and nitrate. In Guangzhou urban areas, nitrate can either be formed
through gas-phase oxidation of $NO_2$ by OH which forms $HNO_3$ and then condenses onto aerosol phase,
or be formed through the hydrolysis of $N_2O_5$, which is formed through reactions between $NO_2$ and
$NO_3$ radical (Yang et al., 2022). The obvious co-increases in nitrate and SOA after sunset indicated
that the decrease of $O_3$ and increase of $NO_2$ consumption had supplied the $NO_3$ and $N_2O_5$ reaction
chains and the increase of ALWC favored the hydrolysis of $N_2O_5$. This was indirectly confirmed when
during winter, despite relatively high concentrations of $O_3$ and $NO_2$ after sunrise compared with other
seasons, nitrate formation was much less prominent due to substantially lower ALWC associated with
lower RH. However, the quick increase of SOA still occurred after sunset despite weak daytime SOA
formation, suggesting that aqueous reactions might play minor roles in nighttime SOA formation that
involve $NO_3$ radical in Guangzhou urban area. The nighttime chemistry that involves $NO_3$ radical
might contribute substantially to organic nitrate formation (Ng et al., 2008;Fry et al., 2009;Rollins et
al., 2012) which would produce the same ions (NO+ and NO2+) with inorganic nitrate due to the
fragmentation of nitrate functionality (-ONO2) under 70 eV electron ionization in the aerosol mass
spectrometer measurements. However, organic nitrate has different fragmentation pattern with that of
inorganic nitrate with previous laboratory studies have shown that the RN=NO+/NO2+ of organic
nitrate is substantially higher than that of inorganic nitrate. Farmer et al. (2010) thus proposed that the
RN variations can be used as an indicator of organic nitrate formations. The Q-ACSM measurements
with unit mass resolution cannot provide accurate measurements of RN due to the resolution limitation
(Allan et al., 2004), however, the resolved RN related to measured nitrate might provide qualitative
constraints on impacts of organic nitrates. The variations of resolved RN as a function of measured
nitrate are shown in Fig.9c, which shows that at high levels of nitrate when inorganic nitrate usually
dominates (Xu et al., 2021), the RN approaches near 2.8 which was close to the inorganic nitrate RN



reported in (Xu et al., 2021), and locates in the range of 1.1-3.5 of inorganic nitrate RN reported in
literatures (Xu et al., 2015b). Diurnal variations of RN under different pollution levels shown in Fig.9d
reveals higher nighttime RN than daytime, and obvious continuous increase of RN after sunset can be
observed for relatively clean and polluted conditions (daily average NR-PM$_1$ of 20-35 μg/m$^3$ to NR-
PM$_1$ of 45-60 μg/m$^3$), suggesting active nighttime organic nitrate formations, which confirmed the
involvement of NO$_3$ radicals in nighttime SOA formations.

**4 Implications for future studies**

In this study, we highlighted the significant roles of SOA in haze formations in Guangzhou urban
area during the entire year and pointed out that for the most prominent and frequent daytime SOA
formations all the year around, both gas-phase photochemistry and aqueous reactions played
significant roles. Therefore, daytime SOA formation was weak in winter when oxidant level and RH
were low, whereas prominent SOA formations were be observed in fall, spring and summer on almost
daily basis. However, how gas-phase and aqueous phase reactions have coordinated to promote the
SOA formation, and the different contributions of gasSOA and aqSOA to SOA formations under
different meteorological conditions and VOCs profiles in different seasons are not clear. In addition,
our results suggested that the coordination of daytime and nighttime SOA formation together had
resulted in highest SOA concentrations in Guangzhou urban area, thus contributed significantly to
severe haze formation. The co-increases of nitrate and SOA after sunrise indicated the significant roles
of nighttime NO$_3$ radical chemistry in promoting haze formations. However, our understanding on
how nighttime chemistry evolved and contributed to secondary aerosols formations in different
seasons is still highly insufficient in this region. Therefore, the precursors and formation pathways of
daytime and nighttime SOA formations and how they coordinated to promote severe haze formations
need further comprehensive investigations to make targeted emission control strategies to continuously
improve air quality in the PRD region. Also, findings of this study have important implications on
future investigations of SOA formation mechanisms in urban areas of southern China that share similar
emission sources and meteorological conditions.
**Data availability**. All data needed are presented in time series of Figures and supplementary Figures,



raw datasets of this study are available from the corresponding author Li Liu (liul@gd121.cn) upon

request.

**Competing interests**. The authors declare that they have no conflict of interest.

**Author Contributions**. YK and LL designed the aerosol experiments. YK conceived and led this research. MMZ and YK wrote the manuscript. MMZ and LL conducted the long-term Q-ACSM measurements. MMZ and YH performed the PMF analysis. HBX, CY, YZ and FL helped maintain and calibrating the Q-ACSM. CL provided meteorological datasets, BL performed the AE33 measurements and post data processing. XJD obtained funding for the continuous aerosol measurements. JCT and WYX provided insights into data analysis, and all authors contributed to revisions of this paper.

## Acknowledgments

This work is supported by the Guangdong Provincial Key Research and Development Program (2020B1111360003); National Natural Science Foundation of China (42175083 and 42105092); Guangdong Basic and Applied Basic Research Foundation (2019A1515110791 and 2019A1515011808); National Key Research and Development Program of China (2019YFCO214605); Science and Technology Innovation Team Plan of Guangdong Meteorological Bureau (GRMCTD202003). The Special Fund Project for Science and Technology Innovation Strategy of Guangdong Province(Grant No.2019B121205004.

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
