# Peer review of "Insights into characteristics and formation mechanisms of secondary organic"

_Atmospheric Chemistry and Physics, 2022_

## Author Comment (AC1)

**General comments:**

The Pearl River Delta (PRD) region is an important area for air quality research. This study provides a year-long observation of aerosol species using aerosol mass spectrometry. The paper's focus is given to secondary organic aerosols which account for about a quarter of submicron particle mass concentration. In general, the paper is well organized and clearly written. Presenting such an extensive observational dataset itself serves as a contribution to the atmospheric chemistry community. I only have a few minor comments on the current paper.

**Responses**: We thank the reviewer for acknowledging our efforts and for all the valuable comments.

**Minor comments**:

**Comment**: On the Q-ACSM data analysis: Previous studies used to have different categories for SOAs. For example, some studies have separated an aq-SOA (aqueous-processing SOA) component. Can the authors provide more information on the assumption and reasons for their choice of SOA categories?

**Response**: We learned from this comment. We did not find any literatures have resolved aqSOA factor on the basis of Q-ACSM measurements. However, we do find the aqSOA factor was resolved from Tof-ACSM measurements which differ with Q-ACSM mainly in resolution of spectrometer, and the aqSOA was characterized by high fraction of m/z 29 (CHO+) and high correlations with sulfate, for example the square of correlation coefficient between sulfate and aq-SOA reached as high as 0.94 in Lei et al. (2021). The square of correlation coefficients between both LOOA and MOOA with sulfate in this study is below 0.5, thus do not support directly if they are related with aqueous phase reactions. To make readers aware of this, we added following discussions in Sect. 2.2: "Note that a aqSOA factor (called aqOOA in these references) was previously resolved using the aerosol mass spectrometer measurements (Sun et al., 2016;Zhao et al., 2019) or time-of flight ACSM measurements (Lei et al., 2021), and the factor was resolved as aqSOA because of its high fraction of m/z 29 (CHO+) and high correlation with sulfate.

Both two resolved SOA factors in this study showed relatively weak correlations with sulfate (Fig.1), do not support directly if they are related with aqueous phase reactions."

**Comment**: Fig.3 & Fig.5: Earlies studies reported fast formation of sulfate at haze episodes, which seems to be different from the measurements shown here. Can the authors show the reasons?

[Figure]

Figure 1. Time series of PM$_{2.5}$ (a) and sulfate (SO$_4$) (b) during autumn of 2020.

**Response**: Thanks for your comment. Sulfate is not the focus of this study, however, this is quite an interesting question and made us noticed that the sulfate evolution during the observations of our study differed much with results shown in previous studies. First, results shown in Fig.3 and Fig.5 do not reflect the formation rate of sulfate.   Sulfate contribute most to submicron aerosol mass during haze episodes of autumn, therefore the time series of sulfate and PM2.5 are shown in Fig.1 as an example. It can be seen that both PM$_{2.5}$ and sulfate rarely show accumulation characteristics that last several days, and shown strong diurnal variations, and fast sulfate formation can be observed in daily scale. This is quite different with results shown in Chen et al. (2021) as shown

in Fig.2, and the results shown in Chen et al. (2021) was observed also at Guangzhou urban area in autumn of 2018. During the observations of Chen et al. (2021), the continuous accumulation of sulfate in some haze episodes is remarkable and totally different with those in observations of this study. Results of these studies highlight the seasonal variations of aerosol chemical compositions might differ much among years. And the reasons behind this might be related with the changes of meteorological conditions and emission conditions. And for the difference between observations of Chen et al. (2021) with this study, the changes of meteorological conditions might play the dominant role, because relative small emissions changes might be expected from 2018 to 2020. To make this point clear to readers, the following discussions was added the Sect 3.2 of the revised manuscript:

"Note that seasonal variations of aerosol chemical compositions might differ much among years due to different meteorological conditions and emissions. For example, the evolution of sulfate during autumn in this study (Fig.S9) have remarkably different

[Figure]

**Figure 2**. Time series of different parameters in Chen et al. (2021).

accumulation characteristics with those observed in autumn of 2018 as shown in Fig.1 of Chen et al. (2021a). Even so, SOA play significant roles in haze formations of Guangzhou urban area in all seasons hold based on results of existing literatures (Zhou

et al., 2020b)."

In addition, sulfate formation mechanism is a hot spot in recent years, however, most investigations on sulfate formations mechanism were done on the North China Plain. Synthesized research on the formation mechanisms of sulfate in the PRD region remain lacking, and might be one of our focuses in next years.

**Comment**: The variations of boundary layer height may affect the interpretation of the formation mechanisms of aerosol species. Can the authors provide more quantitative results on its influence?

**Response**: We agree with the reviewer that the variations of boundary layer height might affect significantly on the interpretation of the formation mechanisms, however, the observations of parameters that relate to planetary boundary layer (PBL) height evolutions is lacking. Reanalysis data of PBL might be available, however, my experiences told me that climatological analysis using reanalysis datasets might be ok, however not accurate enough for resolving hourly scale processes such as SOA formation within several hours. Therefore, in the data analysis of SOA formation, we only tried to qualitatively interpret the results and give clues on future studies. We thought about using CO to scale the observations as done in other studies, however, we found that only small decrease was observed in CO although other parameter such as POA is decreasing substantially, demonstrating that the CO variations reflect poorly PBL evolutions in strong CO source regions.

Chen, W., Ye, Y., Hu, W., Zhou, H., Pan, T., Wang, Y., Song, W., Song, Q., Ye, C., Wang, C., Wang, B., Huang, S., Yuan, B., Zhu, M., Lian, X., Zhang, G., Bi, X., Jiang, F., Liu, J., Canonaco, F., Prevot, A. S. H., Shao, M., and Wang, X.: Real-Time Characterization of Aerosol Compositions, Sources, and Aging Processes in Guangzhou During PRIDE-GBA 2018 Campaign, Journal of Geophysical Research: Atmospheres, 126, e2021JD035114, https://doi.org/10.1029/2021JD035114, 2021.
Lei, L., Sun, Y., Ouyang, B., Qiu, Y., Xie, C., Tang, G., Zhou, W., He, Y., Wang, Q., Cheng, X., Fu, P., and Wang, Z.: Vertical Distributions of Primary and Secondary Aerosols in Urban Boundary Layer: Insights into Sources, Chemistry, and Interaction with Meteorology, Environmental science & technology, 55, 4542-4552, 10.1021/acs.est.1c00479, 2021.

---

## Author Comment (AC2)

**Reviewer #2**

**General comments**:

This study reported a year-long AMS measurement data for NR-PM1 in Guangzhou urban area of the PRD, China, and then discussed the possible formation pathways of SOA and its significant role on haze occurrences in the region. Indeed, long-term AMS data was relatively scarce and thus valuable for better understanding the source contribution and formation pathways of aerosols in the atmosphere. This paper also presented some interesting scientific findings, e.g., haze formation mechanisms differed much among distinct seasons, significant roles of the coordination of gas-phase photochemistry and subsequent aqueous-phase reactions in quick daytime SOA formations, etc. However, in my opinion, several parts of the manuscript were not well organized and clearly described.

**Response**: We thank the reviewer for all the valuable comments and suggestions, which helped us to further improve the readability of our manuscript.

**Detailed comments**

**Comment**: L46-47, "reaching about 1.7 times that of primary organic aerosols …", grammar error.

**Response**: Revised as "reached about"

**Comment**: Introduction: should be entirely revised. Although long-term AMS data is valuable, it is not a scientific goal. The authors should more focus on the present understanding and deficiency of SOA formation, and how the long-term AMS data can help understanding it. The authors also need to summarize the main conclusions of previous long-term AMS studies, not just list these references, and clarify the advantage of this study.

**Response**: Many thanks for this valuable suggestion. We have revised the introduction as suggested by the reviewer:

[revised manuscript text omitted]

**Comment**: L123, "from which black carbon (BC) mass concentrations in winter and early spring", this sentence is confusing.

**Response**: Revised as "from which optically equivalent black carbon (BC) mass concentrations in winter and early spring were calculated."

**Comment**: Q-ACSM data analysis: this section was too redundant, I suggest to move some detailed analysis to supplement.

**Response**: Thanks for your suggestion, we discussed about this part and insist this part should be in the manuscript. Reasons are listed below: (1) PMF analysis of OA spectra is the key part because the identification of OA factors are fundamental for analysis in this study; (2) This is the first paper that resolved OA factors during entire year in the PRD region, direct presentations of detailed discussions about the usage of PMF tool and scientific considerations behind factor determination are important for readers, especially for those are fresh (The first author is fresh two years ago and suffered from learning how to discuss and determine PMF results).

**Comment**: 3.1: L229, L234, etc. In this section, the PBL and rainy conditions were repeatedly referred to explain the seasonal variations of PM1 and chemical componets. Could you provide detailed data about PBL and precipitation for it?

**Response**: Many thanks, we do not have the PBL data during the observations. The discussions are mainly associated with general seasonal variations of PBL in this region which is discussed in Yang et al. (2013), and this reference is added in the revised manuscript. The time series of precipitation is shown in the following Figure, and added as Fig.S9 in the revised supplement.

[Figure]

Time series of precipitation during the observations

**Comment**: L249-250: grammar error.

**Response**: revised as "Nitrate in summer and fall was relatively lower in summer and fall"

**Comment**: L259-261, why are there more traffic emissions in winter? Please provide more evidences.

**Response**: We re-checked the ratio between HOA/COA and didn't find higher HOA/COA ratio during winter, therefore our previous speculation is quite weak and we deleted this sentence.

**Comment**: L265-267, this sentence is confusing, please clarify it.

**Response**: This sentence was revised as:

"Overall, COA contributed about 19% of OA during the whole year which was close to that of HOA (18%). However, the contributions of COA and HOA to total OA differ much among seasons. The contributions of COA to OA were higher than that of HOA during warm months and lower than that of HOA in relatively cold months especially in winter."

**Comment**: 3.2 Significant contributions of secondary organic aerosols to haze formations in all seasons: it seems that the authors try to highlight the role of SOA in haze formations, however, I think it is overstated, at least in the section title. For example, in L286-299, the contribution of OA in generally decreased when PM1 concentration increased in fall. Similar trend of SOA could be found in other seasons (Figure 5).

**Response**: Yes, we want to highlight important roles of SOA in haze formations in all seasons. Haze formations mainly refer to absolute increase/accumulation of aerosol mass, the contribution of OA might decrease in some seasons, however, contribution of SOA generally hold, demonstrating significant contribution of SOA increase to submicron aerosol mass increase as shown in left panels of Fig.5.

**Comment**: L431, 436, NO+ and NO2+ were not in right format.
**Response**: revised

Yang, D., Li, C., Lau, A. K. H., and Li, Y.: Long-term measurement of daytime atmospheric mixing layer height over Hong Kong, Journal of Geophysical Research, 118, 2422-2433, 2013.

---

## Author Response (AR2)

**General comments**:

Thank you for submitting your response to reviewers' comments and revising the paper accordingly. I have reviewed your response and believe you have adequately addressed the comments. However, there are still some edits that need to be made before the manuscript is accepted for publication. Please see the detailed list below.

**Response**: Thank you handling our manuscript and went through very carefully our manuscript, we appreciate this.

Best wishes!

**Specific comments:**

**Comment**: I actually think L46 of abstract is correct if "reaching" is used

**Response**: We agree, revised.

**Comment**: L53: change to "on a daily basis, suggesting important roles of photochemistry in SOA formations"

**Response**: Revised.

Comment: L60-61 change to "This was further confirmed by continuous increase of NO+/NO2+ fragment ratio after sunset which is indicative of formation of particulate organic nitrates".

**Response**: Revised.

**Comment**: L62: "···our understanding of"

**Response**: Revised.

**Comment**: L72: "··· number of studies show.."

**Response**: Revised.

**Comment**: L84: "··· are an active research area of interest in atmospheric chemistry in the recent ten years since significant contributions of SOA to atmospheric aerosol mass have been recognized (Zhang et 85 al., 2007;Jimenez et al., 2009). However, SOA formation is quite complex due to varying precursors, oxidants and formation pathways under different emission characteristics and meteorological conditions."

**Response**: Revised.

Comment: L92Change to "Both field measurements and laboratory studies are needed in investigating detailed SOA formation mechanisms in different regions. Field measurements provide insights into key oxidants and formation pathways under ambient conditions, thus information from field measurements are important for both designing laboratory experiments and targeting emission control strategies."

**Response**: Revised.

**Comment**: L106: "Using these techniques···"

**Response**: Revised.

**Comment**: L107 Change to ". Su et al (2020) found that⋯"
**Response**: Many studies found this phenomena, and Su et al (2020) concluded this. To make this clearer, this sentence was revised as: ", and many studies found that⋯"

**Comment**: L127: delete "and the PRD region importance of long-term measurements"
**Response**: Revised.

**Comment:** L143-144: Change to "however long-term aerosol spectrometer measurements that help characterizing OA sources and SOA formation mechanisms in this region remain lacking"
**Response**: Revised.

**Comment**: L182-183: Change 'peaked' to "peaking"
**Response**: Revised.

**Comment**: L188: "⋯ was not well separated from cooking-related ⋯"
**Response**: Revised.

**Comment**: L194: "⋯urban areas, POA was mainly composed of HOA (which is mostly associated with traffic emissions) and COA, while SOA could be resolved⋯"
**Response**: Revised.

**Comment**: L198 "⋯ ranging from 0.1 to 0.5. Furthermore, we constrained the HOA and COA profiles with HOA and COA profiles reported in Liu et al. (2022) as priories considering ⋯"
**Response**: Revised.

**Comment**: L203: delete "⋯, more details about the method please refer to Liu et al. (2022)"
**Response**: Revised.

**Comment:** L204: "⋯datasets. For example, ⋯."
**Response**: Revised.

**Comment**: L261: "⋯.and therefore, do not support directly aqueous phase SOA formation".
**Response**: Revised.

**Comment**: l277 "⋯ and possibly⋯"
**Response**: Revised.

**Comment**: Figure 6- please add (a), (b), etc to each panel and refer to these in the caption.
**Response**: Revised accordingly.

**Comment**: L377-378: Consider changing "mattered more" to "⋯ was more significant than".

**Response**: Revised.

**Comment**: L382: "··· from those observed.."
**Response**: Revised.

**Comment:** Unclear what you mean here, but if the point is to say the conclusions about SOA are consistent with other studies, change to "However, our conclusions about SOA playing significant roles in haze formations inf Guangzhou urban area during all seasons are consistent among existing literature (e.g., Zhou et al., 2020)."
**Response**: Many thanks! Your understanding is what we want to deliver, and corrected as you suggested.